# Artificial Intelligence: A New Diagnostic Software in Dentistry: A Preliminary Performance Diagnostic Study

**DOI:** 10.3390/ijerph19031728

**Published:** 2022-02-02

**Authors:** Francesca De Angelis, Nicola Pranno, Alessio Franchina, Stefano Di Carlo, Edoardo Brauner, Agnese Ferri, Gerardo Pellegrino, Emma Grecchi, Funda Goker, Luigi Vito Stefanelli

**Affiliations:** 1Department of Oral and Maxillofacial Sciences, Sapienza University of Rome, Via Caserta 6, 00161 Rome, Italy; nicola.pranno@uniroma1.it (N.P.); stefano.dicarlo@uniroma1.it (S.D.C.); edoardo.brauner@uniroma1.it (E.B.); gigistef@libero.it (L.V.S.); 2Private Practice, Via Legione Gallieno 44, 36100 Vicenza, Italy; alessiofranchina@icloud.com; 3Oraland Maxillofacial Surgery Division, DIBINEM, University of Bologna, 40125 Bologna, Italy; agnese.ferri@ymail.com (A.F.); gerardo.pellegrino2@unibo.it (G.P.); 4Chirurgiche ed Odontoiatriatriche, Dipartimento di Scienze Biomediche, University of Milan, Via Della Commenda 9, 20122 Milano, Italy; emma.grecchi@gmail.com (E.G.); funda.goker@unimi.it (F.G.)

**Keywords:** artificial intelligence, machine learning, digital dentistry, dental radiology, accuracy

## Abstract

Background: Artificial intelligence (AI) has taken hold in public health because more and more people are looking to make a diagnosis using technology that allows them to work faster and more accurately, reducing costs and the number of medical errors. Methods: In the present study, 120 panoramic X-rays (OPGs) were randomly selected from the Department of Oral and Maxillofacial Sciences of Sapienza University of Rome, Italy. The OPGs were acquired and analyzed using Apox, which takes a panoramic X-rayand automatically returns the dental formula, the presence of dental implants, prosthetic crowns, fillings and root remnants. A descriptive analysis was performed presenting the categorical variables as absolute and relative frequencies. Results: In total, the number of true positive (TP) values was 2.195 (19.06%); true negative (TN), 8.908 (77.34%); false positive (FP), 132 (1.15%); and false negative (FN), 283 (2.46%). The overall sensitivity was 0.89, while the overall specificity was 0.98. Conclusions: The present study shows the latest achievements in dentistry, analyzing the application and credibility of a new diagnostic method to improve the work of dentists and the patients’ care.

## 1. Introduction

The use of artificial intelligence in healthcare and dentistry is booming. In recent years, traditional dentistry has increasingly transformed into digital dentistry due to the introduction of different software applied to different machines used in the medical field [1]. Artificial intelligence (AI) has taken hold in public healthcare as more and more people try to make a diagnosis using technology that allows them to work faster and more accurately, reducing costs and the number of medical errors [2]. The future is based on an ever-increasing quest for innovation and development to achieve high quality in the treatment of patients. The term AI was coined by John McCarthy [3] in 1989 referring to machines that imitated human behavior and knowledge. This capability has been enriched by sequences of algorithms through the development and improvement of hardware. The question often asked by clinicians is as follows: “Will artificial intelligence be able to replace the role of the clinician in the diagnosis of diseases?” This question has led to a great deal of interest in the scientific literature in the field of technology, especially in AI applied to various medical disciplines [4]. In 1950, Alan Turing was the first to ask whether machines were capable of thinking [5]. In the field of clinical medicine, a large number of AI models have been developed to assess the risk or presence of diseases and thus the diagnosis and assessment of a prognosis [6,7]. The healthcare sector continues to evolve, and especially in the last five years, AI and machine learning have become more and more popular with very substantial investments both in the private sector and in the public sector. AI is a technology whereby humans, by means of complex software algorithms, manage to transfer to a machine tasks that would normally be solved exclusively by the human mind; indeed, very often they are even tasks that would have to be carried out by entire teams of people with different skills and experience. However, the use of this basic concept of AI has very different feedback depending on the purpose for which it is used and especially for the specific field of use, such as healthcare. AI applications in healthcare are becoming more common for the automation of certain tasks, especially for diagnostics to support doctors’ decisions. AI applied to the huge amount of data produced by healthcare facilities allows enormous benefits with a variety of possibilities for contributions, for example predictive, more targeted and customized healthcare prevention; better and more accurate detection of symptoms; automated use of analysis results (images, laboratory analysis, etc.); formulation of treatment plans or customized protocols; and facilitation of the coordination of care teams. AI thus enables us to enter a new era of extremely early diagnosis, through the search and detection of pre-symptoms or predisposition to contract a given disease. The biggest field of application for AI is certainly radiology, as it uses digitally encoded images that can easily be translated into computer language [8].

In diagnostic imaging, therefore, the advantages of AI are and will be very significant [9] because with the automatic reading of the images radiologists will be able to concentrate only on the interpretation of complex pathologies and/or orient themselves towards interventional radiology. This is already creating a huge gap in the time taken to process examination results between a hospital that adopts AI techniques for the interpretation of examinations and those that rely solely on traditional examination, with the obvious repercussion also on waiting lists. In the field of dentistry, preclinical studies have been conducted on diagnostic models to accurately visualize root morphology [10]. AI has been used to improve image interpretation in dental radiology [11]. The collaboration between clinician and engineer is essential because the clinical experience allows finding anatomical landmarks while the engineer’s experience allows using the software in the most correct way by managing the information set [12].

The applications of AI in the field of dentistry can vary according to need, from dental emergencies to prosthetic planning. AI allows machines to learn from experience, adapt to new inputs and perform tasks in a human-like manner [13,14]. By using large amounts of data, including diagnostic results, treatments and outcomes, AI will be able to measure the effectiveness of different treatment modalities associated with specific symptoms and anatomical conditions and improve the quality of standardization processes [15]. The primary objective of this preliminary study was to evaluate the accuracy measures of a diagnostic tool based on artificial intelligence and machine learning. The secondary objective was to analyze the accuracy of this tool for the diagnosis of different dental diseases or conditions.

## 2. Materials and Methods

The study was designed as a cross-sectional performance diagnostic study.

### 2.1. Data Selection

In the present study, 120 panoramic X-rays (OPGs) were randomly selected from the Department of Oral and Maxillofacial Sciences of Sapienza University of Rome, Italy. The OPGs could be jpeg (JPEG) or dicom (DICOM or DCM). The OPGs were acquired using Apox (Promaton, Amsterdam, the Netherlands) which takes a panoramic X-ray and turns it into clinical insights. Currently, Apox can identify most dental structures with the same accuracy as a dentist. The current version of Apox helps clinicians to automatically file the most common dental structures, saving them time. Apox is offered through an application programming interface (API) and integrates easily into existing dental software systems. The collected data were anonymized prior to analysis. This study has been conducted in accordance with the code of ethics of the World Medical Association (Declaration of Helsinki).

### 2.2. Data Annotation and Software Procedures

The OPGs were analyzed by two experienced clinicians (FDA, LVS), and if differences were found, a third experienced clinician (AF) made the decision.

Each of the clinicians was trained in the analysis of 50 orthopanoramic images before starting the study. All the three clinicians used a standardized protocol for the data annotation process.

This study assessed the dental formula, paying particular attention to the presence or absence of the tooth element in the correct position. Each OPG was uploaded into the Apox software using the drag and drop tool. The software analyzes the OPGs uploaded and automatically returns the following outcomes after one minute (Figure 1):-dental formula;-dental implants;-prosthetic crown;-fillings;-root remnants;-root canal treatments.

**Figure 1 ijerph-19-01728-f001:**
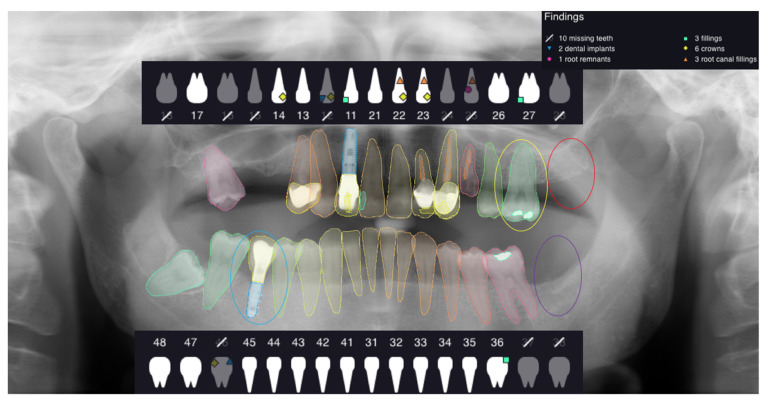
The figure shows an example of analyzed OPG and the outcomes given by the software.

The outcomes obtained were analyzed by the two experienced clinicians, and one of the following values was assigned for each data item analyzed: true positive (TP), true negative (TN), false positive (FP), false negative (FN). The inter-rater reliability was calculated using Cohen’s kappa coefficient.

To the correct identification of the analyzed element in the correct position by the software was assigned the value of TP.

For the absence of the same analyzed elements in the correct position, the value of TN was given; in the cases of software error, the value of FN was assigned when the presence of the analyzed element was not detected by the software, and the value of FP was assigned in the case in which the software indicated the presence of one of the analyzed elements that did not exist (Figure 2, Figure 3, Figure 4 and Figure 5).

All values were reported in an Excel table (Figure 6).

### 2.3. Statistical Analysis

A descriptive analysis was performed presenting the categorical variables as absolute and relative frequencies. The following diagnostic accuracy measures were determined: sensitivity, specificity, positive predictive value (PPV), negative predictive value (NPV), positive likelihood ratio (LR+) and negative likelihood ratio (LR−).

Receiver operating characteristic (ROC) curve analysis was performed. Then, the areas under the curve (AUCs), derived indexes of accuracy, were calculated to describe the overall diagnostic performance of the AI software and to compare the diagnostic accuracy for each group variable (presence or absence of teeth, implants, crowns, fillings, residual roots and root canal treatments). The overall ROC curve and it’s AUC were calculated considering the covariate identified as the pathology/condition group. The nonparametric method of Delong et al., (1988) for the calculation of the standard error of the AUC sand of the difference between the different AUCs was used.

Sensitivity was defined as the proportion of TP in a total group of subjects with the specified condition: TP/(TP + FN). Sensitivity estimated the probability of obtaining a positive test result in case the condition was present. Hence, it relates to the ability of the test to recognize the presence of the specified condition.

Specificity was defined as the proportion of TN in a total group of subjects without the condition: TN/(TN + FP). It estimated the probability of obtaining a negative test result in case the condition was absent. Therefore, it relates to the ability of a diagnostic procedure to recognize the absence of a condition.

PPV described the probability of the specified condition being present in case of a positive result. Therefore, a PPV represents a proportion of true positive test result in a total group of subjects with a positive result: TP/(TP + FP).

NPV described the probability of absence of the condition in case of a negative test result. An NPV is defined as a proportion of subjects without the condition and with a negative test result in a total group of subjects with a negative test result: TN/(TN + FN).

LR+ told us how many times more likely a positive test result was to occur in a case with the presence of the specified condition than in a case without the condition. The farther LR+ was from 1, the stronger the evidence for the presence of the condition. LR+ was calculated according to the following formula: LR+ = sensitivity/(1–specificity). LR− represented the ratio of the probability that a negative result would occur in case the condition was present to the probability that the same result would occur in case the condition was absent. Therefore, LR− told us how much less likely the negative test result was to occur in a subject with the condition than in a subject without the condition. LR− was calculated according to the following formula: LR− = (1–sensitivity)/specificity. The lower the LR−, the stronger was the evidence for the absence of the condition.

Diagnostic effectiveness is a global measure of diagnostic accuracy and is affected by disease prevalence. It was expressed as a proportion of correctly classified subjects (TP + TN) among all subjects (TP + TN + FP + FN).

ROC curves were constructed plotting 1–specificity on the *x*-axis and sensitivity on the *y*-axis. The shape of the ROC curve and the AUC estimated how high the discriminative power of the AI test was. The closer the curve was located to the upper-left corner and the larger the AUC, the better the test could discriminate between the absence or the presence of studied conditions. The AUC can have any value between 0 and 1 and is a good indicator of the goodness of the test.

The significance level was set at 0.05. All analyses were performed using Stata version 15 (Stata Corp. LP, College Station, TX, USA) by a blind operator.

## 3. Results

The Cohen’s kappa coefficient was 1.0, showing a perfect level of agreement between the data collectors.

In total, there were 2195 TP (19.06%), 8908 TN (77.34%), 132 FP (1.15%) and 283 FN (2.46%) values.

The overall sensitivity was 0.89, while the overall specificity was 0.98. The PPV was 0.94 and the NPV was 0.97. The LR + was 44 and the LR− was 0.001. The diagnostic effectiveness had a value of 0.96.

TP, TN, FP and FN values calculated for each group (teeth, implants, crowns, fillings, residual roots and root canal treatments) are reported in Table 1.

Regarding the diagnosis of absence or presence of teeth and implants, the sensitivity was0.95 and 0.84, specificity was 0.90 and 0.99, PPV was 0.95 and 0.87, NPV was 0.88 and 0.99, LR+ was 9.5 and 84, LR− was 0.055 and 0.16 and diagnostic effectiveness was 0.93 and 0.94, respectively.

Regarding the detection of crowns and fillings, the sensitivity was 0.93 and 0.61, specificity was 0.99 and 0.99, PPV was 0.95 and 0.74, NPV was 0.98 and 0.99, LR+ was 93 and 61, LR− was 0.07 and 0.39 and diagnostic effectiveness was 0.98 and 0.99, respectively. In determining the presence or absence of residual roots and root canal treatments, the sensitivity was 0.68 and 0.83, specificity was 0.98 and 0.99, PPV was 0.91 and 0.92, NPV was 0.92 and 0.97, LR+ was 34 and 83, and LR− was 0.33 and 0.17 and diagnostic effectiveness was 0.92 and 0.87, respectively.

The area under the ROC curve (AUC), controlling for the variable group as covariate, was 0.94 (Figure 7).

The AUCs calculated for the ROC curves of each group variable are presented in Table 2.

The AUC calculated for the root residual group (0.80, 95% CI 0.70–0.90) was minor and was significantly lower with respect to those of teeth (0.92, 95% CI 0.91–0.94) and prosthetic crowns (0.96, 95% CI 0.94–0.97). The latter had the biggest value and was significantly higher with respect to groups of fillings (0.83, 95% CI 0.81–0.85), residual roots (0.80, 95% CI 0.70–0.90) and root canal treatments (0.90, 95% CI 0.88–0.93). ROC curves of all variables are presented in Figure 8.

## 4. Discussion

In recent years, we have been looking for a correct and standardized way to make a diagnosis of the most common dental diseases using two-dimensional (2D) X-rays and taking into account the three-dimensional (3D) images derived from the CT cone beam.

A study in recent literature concerning the use of AI systems applied to dental and maxillofacial radiology revealed the presence only of in vitro studies, skull studies and studies on anatomical models [16].

No clinical studies have been performed to verify the actual application of these systems on the patient.

The aims of the present study were the evaluation of the accuracy measures of a diagnostic tool based on artificial intelligence machine learning and the analysis of this tool’s accuracy for the diagnosis of different dental pathologies or conditions.

Automated tooth detection, classification and numbering are fields of great interest and can simplify the digital compilation of dental charts.

The results reported in this study showed high values of sensitivity and specificity for all the variables analyzed and they represent an important starting point for further developments of this kind of software.

Miki et al., in their practice have developed an algorithm to classify the type of tooth based on the three-dimensional image of the CBCT [17]. The application of artificial intelligence can contribute to the automatic identification of missing teeth for the diagnosis and planning of dental implants or prosthetic treatments. Other authors such as Ghazvinian Zanjani and Kim have developed algorithms for the digitization of three-dimensional dental surfaces, obtaining a high accuracy of segmentation [18,19].

These studies have accelerated the digital workflow and reduced human error. In recent years, AI has found application in a large number of medical fields [20,21].

In the past, artificial intelligence had not found great application in dentistry, but in recent years there has been a growing interest from many dentists who have started to want to apply it to patients thanks to increasingly advanced software.

We have examined the performance of an AI software (Promaton) to evaluate the ability to recognize the number of teeth, roots, implants and fillings by evaluating the margin of error, and we found that even if the results are promising, there are still some errors that need to be eliminated.

In particular, we found the best values (sensitivity >0.9) for the correct detection of the presence of the teeth and crowns and the worst results (sensitivity <0.7) for the correct detection of fillings and residual roots. The low values for the correct detection of the presence of fillings could be explained by the majority of the fillings being radiolucent. High values (specificity >0.9) were found for a lack of detection of all the analyzed subjects.

In a future scenario, AI will be the potential first screening system, especially for public medicine as well as for dental medicine. To be sure that no tooth or implant is not recognized, it is, in the authors’ opinion, a primary requisite to routinely introduce AI in the diagnostic workflow.

Most of the artificial intelligence algorithms studied over the years by various authors have been designed to address various clinical situations in various fields of medicine. The future project they are expected to achieve is the creation of an integrated digital workflow system that exploits artificial intelligence for the dental sector [22,23,24].

In fact, after entering patient data, 2D and 3D radiological images and intraoral and facial scans, the system could automatically provide patient analysis and a treatment plan. In this way, personalized dental medicine could be developed with an individualized diagnosis and a prediction of the treatment outcome. The dentist will thus obtain a large database updated in real time, which will undoubtedly improve the dentist’s work and patient care [12].

The algorithms based on AI can not replace the dentist’s role in the diagnosis of dental pathologies, but surely dentists could benefit from the use of artificial intelligence to receive a second opinion in a few seconds and help patients [25,26].

AI could also be useful in cases of responsibility in diagnosis. Dentistry is moving towards a new era of medicine based on robot-assisted AI. AI has not yet been fully introduced into dental research, so it remains a promising challenge. In the future, we expect to obtain easier treatment plans with a reduction in errors and an increase in the effectiveness of the healthcare system through the use of AI [27]. Dentists will need to acquire familiarity with digital systems, including the ability to interact between digital robots and humans [28]. More clinical studies should be funded in the future to facilitate the testing of this new technology in clinical practice [29,30].

Because of the absence of studies regarding this specific topic, the present research was designed as a preliminary performance diagnostic study. For this reason, it has to be considered that it could not guarantee robust results and that the internal validity of the study was inevitably affected.

## 5. Conclusions

The use of artificial intelligence has great potential in dentistry. The application of artificial intelligence in dentistry is considered very useful for analyzing dental X-rays for early diagnosis. In fact, dentistry is a field that greatly needs the support of technology because it still lacks a standard level of quality: in fact, if patients go to different dentists, they might get different opinions. All this will lead to a unique treatment plan for each patient using software assistance, from planning to treatment and follow-up. This software will have the ability to learn automatically and therefore improve its performance from time to time. AI, therefore, helps clinicians to improve the quality of their diagnoses, save time and increase profitability.

In order to use the advantages of AI correctly, it is important to dose the use of these tools with intelligence, objectivity and common sense, with an appropriate learning curve.

## Figures and Tables

**Figure 2 ijerph-19-01728-f002:**
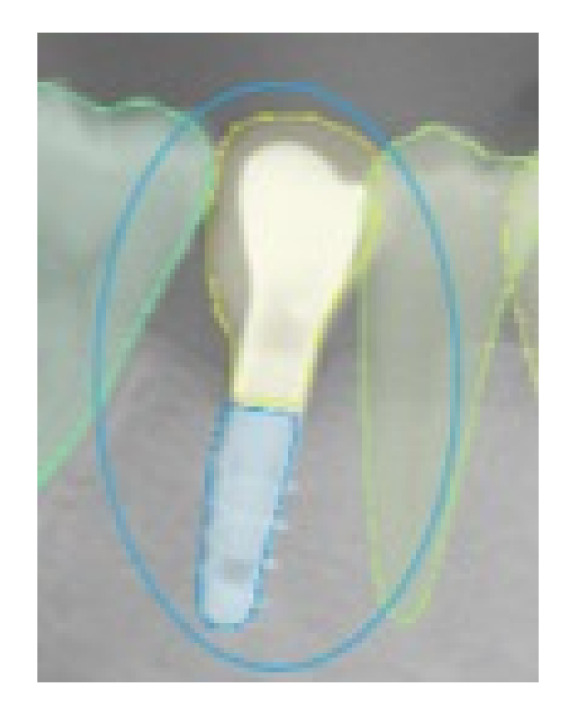
This selected view of Figure 1 shows the correct detection in the right position of an implant. The assigned value was TP.

**Figure 3 ijerph-19-01728-f003:**
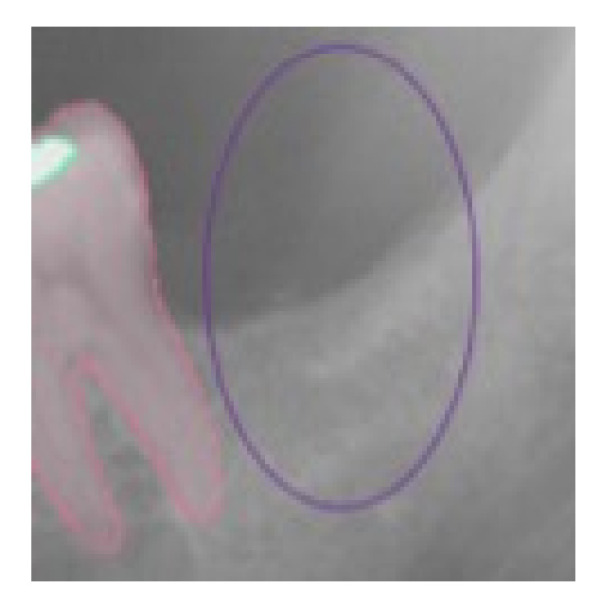
This selected view of Figure 1 shows the correct detection in the right position of an absent tooth in position 3.7. The assigned value was TN.

**Figure 4 ijerph-19-01728-f004:**
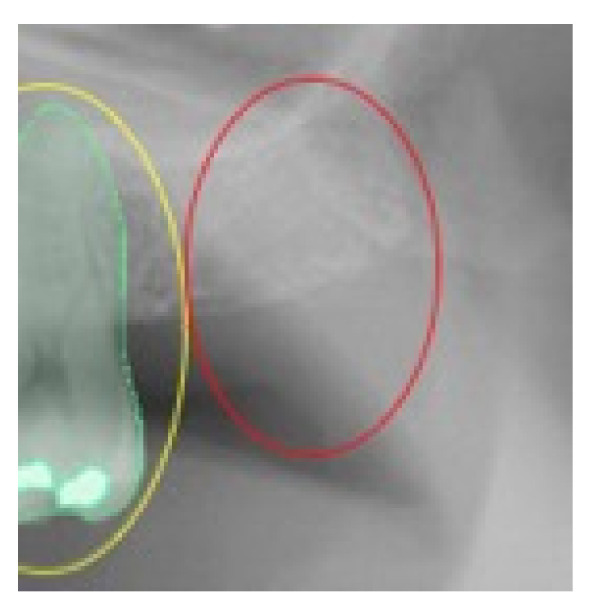
This selected view of Figure 1 shows the incorrect detection in the right position of the tooth 2.7. The assigned value was FP.

**Figure 5 ijerph-19-01728-f005:**
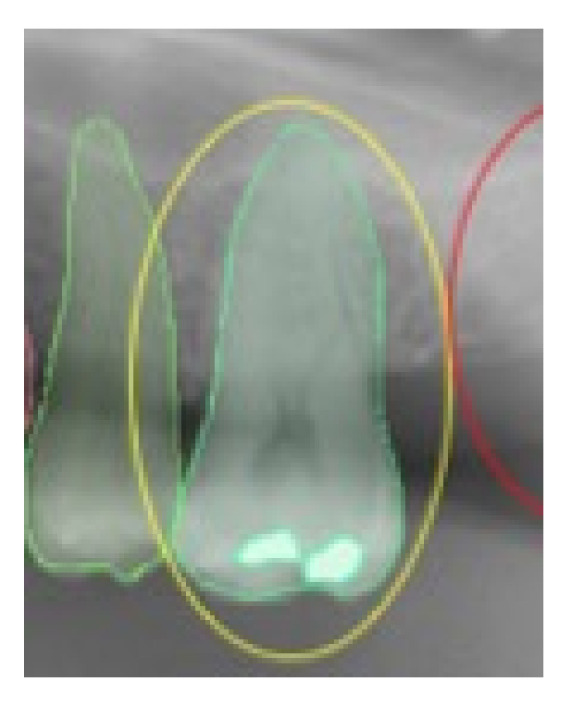
This selected view of Figure 1 shows the missing detection of the tooth filling in position 2.6. The assigned value was FN.

**Figure 6 ijerph-19-01728-f006:**
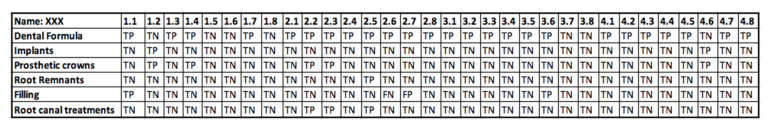
An example of data collected from an OPG using an Excel table.

**Figure 7 ijerph-19-01728-f007:**
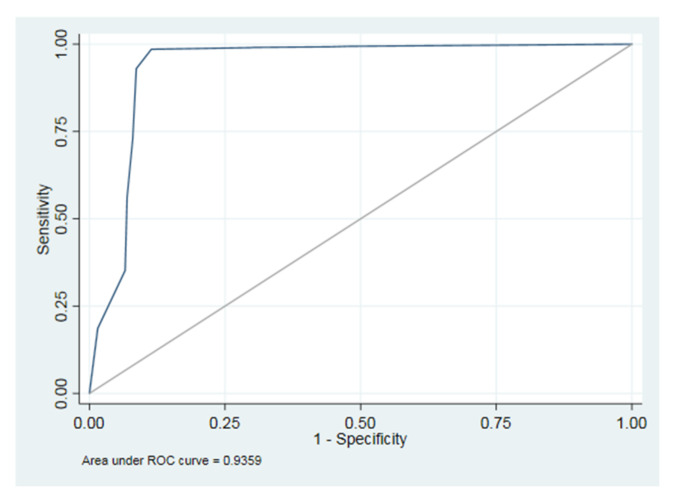
The area under the ROC curve (AUC).

**Figure 8 ijerph-19-01728-f008:**
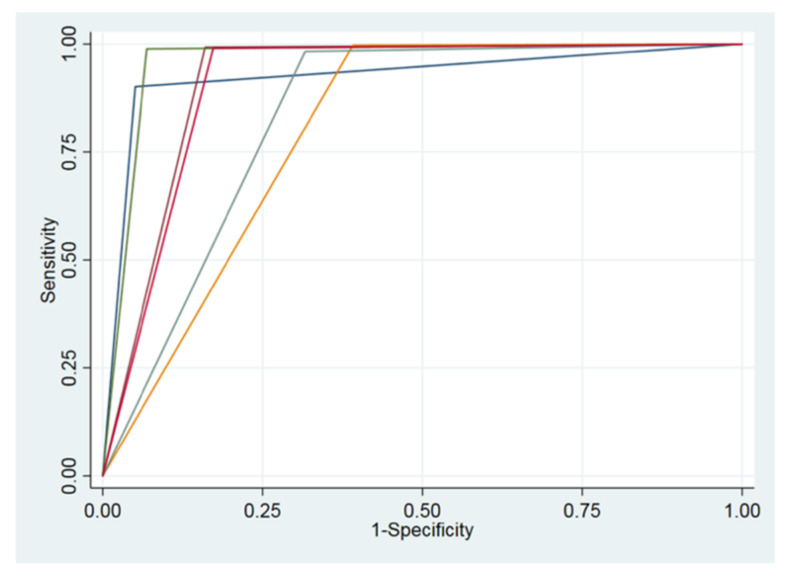
ROC curves of all variables.

**Table 1 ijerph-19-01728-t001:** TP, TN, FP and FN values calculated for each group (teeth, implants, crowns, fillings, residual roots and root canal treatments).

	Freq.	Percent.	Cum.
TP	182	9.48	9.48
TN	1684	87.71	97.19
FP	16	0.83	98.02
FN	38	1.98	100.00
Total	1920	100.00	

**Table 2 ijerph-19-01728-t002:** The AUCs calculated for the ROC curves of each group variable.

Group	Obs	ROC	Std. Err.	[95% Conf. Interval]
0	1920	0.9253	0.0070	0.91162 0.93901
1	1919	0.9164	0.0184	0.88027 0.95259
2	1920	0.9600	0.0067	0.94694 0.97308
3	1920	0.8030	0.0520	0.70106 0.90500
4	1919	0.8332	0.0118	0.81005 0.85643
5	1920	0.9089	0.0128	0.88379 0.93407

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
