# Peer review of "Artificial Intelligence: A New Diagnostic Software in Dentistry: A Preliminary Performance Diagnostic Study"

_ijerph, 2022, doi:10.3390/ijerph19031728_

Round 1
Reviewer 1 Report
This is an interesting paper comparing clinician interpretations of OPG's against AI supported software.
The introduction could be re-written to focus more on AI uses in healthcare briefly with a focus on dentistry.
The methods section could be made more succinct. Have any power calculations been made & if not this could be included in limitations.
Although the method of clinician calibration was mentioned, this should be further explained, for example how were the outcomes of the training on the 50 OPG's considered in terms of equivalency.
The discussion could consider each group of result individually & could further elaborate on each including the potential clinical significance. For example, the detection of a non-existent tooth would likely have larger clinical consequences compared to the non-detection of a restoration.
A clear declaration of any conflicts of interest needs to be included, for example was this study in any way supported by the makers of Apox software or have the authors any financial interests?
The conclusion needs to be re-written as it currently fails to clearly address the aim of the study.
Author Response
We are grateful for your consideration of this manuscript, and we have very much appreciated your suggestions. The paper has been revised inlight of the reviewers' suggestions and comments. We hope that the revision will meet your approval.
RESPONSES TO REVIEWER 1
1.The introduction could be re-written to focus more on AI uses in healthcare briefly with a focus on dentistry.
Our response:theintroduction has been re-written.
2.The methods section could be made more succinct. Have any power calculations been made & if not this could be included in limitations.
Our response: No sample size calculation was performed because of the preliminary nature of the study. Further studies could be based on these results to establish their sample size. As you suggested, we added this limitation in the Discussion paragraph.
3.Although the method of clinician calibration was mentioned, this should be further explained, for example how were the outcomes of the training on the 50 OPG's considered in terms of equivalency.
Our response: Thank you for this question that offers us the opportunity to specify the level of agreement between the data collectors. We added in the text the Cohen’s kappa coefficient result, showing that the inter-rater reliability was 100%.
4.The discussion could consider each group of result individually & could further elaborate on each including the potential clinical significance. For example, the detection of a non-existent tooth would likely have larger clinical consequences compared to the non-detection of a restoration.
Our response:The discussion session has been enriched with some considerations about your suggestion
5.A clear declaration of any conflicts of interest needs to be included, for example was this study in any way supported by the makers of Apox software or have the authors any financial interests?
Our response: A statement regarding the absence of any conflict of interest has been added at the end of the manuscript
Our response:
6.The conclusion needs to be re-written as it currently fails to clearly address the aim of the study.
Our response:the conclusion has been re-written.
Reviewer 2 Report
Author and Editor
Thank you for your work. Good topic, but a number of points need clarifying before publish. These are given below.
Minor
Abstract:
No need to write the software.
Materials and methods:
What were the inter-rater reliability of the examiners?
What is the meaning of API in line no 93? This terminology mentioned previously or not?
Results
Some error occurs after using software. So, how many % of correct accuracy will be achieved in the present study after using this software?
Conclusion
Why citation is needed in the conclusion? It should move to the discussion part.
Author Response
We are grateful for your consideration of this manuscript, and we have very much appreciated your suggestions. The paper has been revised inlight of the reviewers' suggestions and comments. We hope that the revision will meet your approval.
1.Abstract:No need to write the software.
Our response: As suggested, the correction has been made.
2.Materials and methods:What were the inter-rater reliability of the examiners?
Our response: Thank you for this question that offers us the opportunity to specify in the text the level of agreement between the data collectors. The inter-rater reliability was 100%, so the Cohen’s kappa coefficient was 1.0.
3.What is the meaning of API in line no 93? This terminology mentioned previously or not?
Our response: As suggested, the meaning of API was added at line n°95.
4.Results: Some error occurs after using software. So, how many % of correct accuracy will be achieved in the present study after using this software?
Our response: The total accuracy was 94%. We better underlined this result in the Results section, to be better identifiable and understandable in the text.
5.Conclusion: Why citation is needed in the conclusion? It should move to the discussion part.
Our response:citation was moved at line n.°248 in the discussion part as you suggested.